# Assessment of Steady and Unsteady Friction Models in the Draining Processes of Hydraulic Installations

Óscar E. Coronado-Hernández [1,*], Ivan Derpich [2], Vicente S. Fuertes-Miquel [3], Jairo R. Coronado-Hernández [4] and Gustavo Gatica [5]

1 Facultad de Ingeniería, Universidad Tecnológica de Bolívar, Cartagena 131001, Colombia
2 Departamento de Ingeniería Industrial, Universidad Santiago de Chile, Santiago de Chile 9170020, Chile; ivan.derpich@usach.cl
3 Departamento de Ingeniería Hidráulica y Medio Ambiente, Universitat Politècnica de València, 46022 Valencia, Spain; vfuertes@upv.es
4 Departamento de Gestión Industrial, Agroindustrial y Operaciones, Universidad de la Costa, Barranquilla 080001, Colombia; jcoronad18@cuc.edu.co
5 Faculty of Engineering, Universidad Andres Bello, Santiago de Chile 7500971, Chile; ggatica@unab.cl
* Correspondence: ocoronado@utb.edu.co; Tel.: +57-301-371-5398

**Abstract:** The study of draining processes without admitting air has been conducted using only steady friction formulations in the implementation of governing equations. However, this hydraulic event involves transitions from laminar to turbulent flow, and vice versa, because of the changes in water velocity. In this sense, this research improves the current mathematical model considering unsteady friction models. An experimental facility composed by a 4.36 m long methacrylate pipe was configured, and measurements of air pocket pressure oscillations were recorded. The mathematical model was performed using steady and unsteady friction models. Comparisons between measured and computed air pocket pressure patterns indicated that unsteady friction models slightly improve the results compared to steady friction models.

**Keywords:** air pocket; draining process; friction factor; transient flow; unsteady





## 1. Introduction

Mathematical models have been proposed for simulating draining and filling operations [1] in water installations. In short time periods, these hydraulic events can cause dangerous pressure surges for filling processes and sudden drops in sub-atmospheric pressure during draining operations, depending on the magnitude of air pocket volumetric changes and the characteristics of water pipelines [2–4]. The complexity of these phenomena involves the study of governing equations for the water column, the polytropic law of entrapped air pockets, and the air–water interface formulation [5,6]. The occurrences of these hydraulic events imply variation of the Reynolds number and the friction factor, since the water velocity changes over time [7].

The majority of developed mathematical models for studying these processes consider a constant friction factor; some of them are as follows: Zhou et al. [4] investigated the effect generated by two entrapped air pockets during a filling process; Wang et al. [8] analyzed the implications of a rapid filling operation in bypass water pipelines; Vasconcelos et al. [8] studied the pressure surges in stormwater tunnels; Izquierdo et al. [9] analyzed the influence of a trapped air pocket in a water installation of irregular profile; Laanearu et al. [10,11] proposed a semi-empirical model for analyzing a draining operation; and Fuertes-Miquel et al. [3] and Coronado-Hernández et al. [12] proposed a mathematical model for studying the emptying operation in water pipelines.

The current models consider a constant friction factor, which can be computed using methodologies such as: (i) The Colebrook–White equation [13], which is based on a

physical approach, and (ii) the empirical formulations of Moody [14], Wood [15], Hazen-Williams [16], and Swamee–Jain [17].

Steady friction models (SFMs) are used to compute the head losses per unit length, and include the relationship between the friction factor, the water velocity, and the internal diameter of pipelines. However, when a transient phenomenon occurs, then unsteady friction models (UFMs) are recommended for evaluating the head losses [18,19], since they relate the convective acceleration term, the local acceleration term, the wave speed, the gravity acceleration, and the variables and parameters of SFMs. UFMs have been implemented for studying transient flow considering the water phase under the scenarios of the closing of regulating valves, the stoppage of pumps, as well as the use of protection devices to control water hammer events [19–21].

Recently, a UFM was implemented for studying a transient event during a rapid filling operation in a vertical pipe [7], showing a good agreement regarding the comparison conducted between the measured and computed air pocket pressure patterns. There are no detailed studies concerning the analysis of UFMs during draining processes without admitting air [22–24].

The main objective of this research was to establish the governing equations to simulate numerical draining processes considering a UFM. A comparison of an STM and a UFM was performed to note differences using these kinds of formulations. In order to select the best one, an experimental facility was configured at the Universitat Politècnica de València, which was composed by a 4.36 m long inclined methacrylate pipe with an internal diameter of 42 mm. During the experiments, absolute air pocket head pressure patterns were measured using a pressure transducer located at the highest point of the water installation.

## 2. Mathematical Model

This section presents the used governing equations to simulate the draining processes in water pipelines without admitting air, which was established by the authors in previous publications [3,12], assuming a constant friction factor. The Moody, Wood, and Hazen–Williams equations were implemented in the mathematical model for simulating draining operations in water pipelines in order to expand the current literature. At the end, the implementation of an unsteady friction model (UFM) for the rigid water column equation was conducted.

### 2.1. Governing Equations

- Rigid water column model (RWCM):

This equation describes the water movement along a pipeline while an emptying process is occurring. The RWCM neglects the elasticity of water and pipe volumetric changes, since these values are negligible compared to the elasticity of the air phase. The equation applied to an emptying column is shown as follows:

$$\frac{dv}{dt} = \frac{p_1^* - p_{atm}^*}{\rho_w L} + g\frac{\Delta z}{L} - gJ_{s/u} - \frac{R_v g A^2 v |v|}{L}, \tag{1}$$

where:

$v$: water velocity, m/s;
$p_1^*$: air pocket absolute pressure, Pa;
$p_{atm}^*$: atmospheric pressure, 101,325 Pa;
$\rho_w$: water density, kg m$^{-3}$;
$L$: length of a water column, m;
$J_{s/u}$: head losses using a steady ($J_s$) or unsteady ($J_u$) friction model, m/m;
$g$: gravitational acceleration, m s$^{-2}$;
$\Delta z$: difference elevation, m;
$D$: internal diameter of a pipe, m;
$R_v$: resistance coefficient of a valve, s$^2$ m$^{-5}$;

$A$: cross-sectional area of a pipe, m$^2$.

- Air–water interface formulation:

The piston model formulation can be used to compute the air–water interface for a pipeline:

$$\frac{dL}{dt} = -v, \tag{2}$$

- Polytropic law of an air pocket:

The behavior of an entrapped air pocket during an emptying process can be simulated with the polytropic law:

$$p_1^* x^k = p_{1,0}^* x_0^k, \tag{3}$$

where:

$x$: air pocket length, m;
$k$: polytropic coefficient;
0: refers to initial conditions.

The algebraic–differential equation system, composed by Equations (1)–(3), is used to evaluate the behavior of hydraulic and thermodynamics variables (water velocity, air pocket pressure, and length of the water column). The initial condition of the systems is: $v(0) = 0$, $p_{1,0}^* = p_{atm}^*$, and $x_0 = L_T - L_0$, where $L_0$ refers to the initial position of the water column and $L_T$ is the total length of the pipe.

*2.2. Steady Friction Model (SFM)*

A steady friction model is used for estimating the head losses per unit length, expressed as:

$$J_s = f\frac{v|v|}{2gD}, \tag{4}$$

where:

$f$: friction factor;
$J_s$: the head losses per unit length in the steady flow regime.

The implementation of Equation (1) is based on the definition of the SFM (see Equation (4)).

The friction factor measures the resistance to flow by a water installation. For a laminar flow, the friction factor is computed as $f = 64/\text{Re}$. For a turbulent flow, physical and empirical approaches have been considered. The following empirical formulations of computing the friction factor are used.

- Moody equation:

This empirical equation was established by Lewis Ferry Moody based on the Moody chart, which relates the friction factor, the Reynolds number, and the relative pipe roughness ($k_s/D$):

$$f = 0.0055\left[1 + \left(20{,}000\frac{k_s}{D} + \frac{10^6}{\text{Re}}\right)^{1/3}\right], \tag{5}$$

$k_s$: absolute pipe roughness, mm;
Re: Reynolds number.

- Wood equation:

This empirical equation was arranged by Donald Wood, which relates:

$$f = 0.094\left(\frac{k_s}{D}\right)^{0.225} + 0.53\left(\frac{k_s}{D}\right) + 88\left(\frac{k_s}{D}\right)^{0.44}\text{Re}^{1.62\left(\frac{k_s}{D}\right)^{0.134}}, \tag{6}$$

- Hazen–Williams equation:

This is the most famous empirical equation that can be used to compute the friction factor, relating the Hazen–Williams coefficient, the internal pipe diameter, the water velocity, and the Reynold number:

$$f = \frac{133.89}{C_{HW}^{1.851} D^{0.017} v^{0.15} \mathrm{Re}^{0.15}},$$ (7)

where:

$C_{HW}$: Hazen–Williams coefficient.

- Swamee–Jain equation:

This equation is a simplified version of the Colebrook–White formulation, which directly computes the friction factor considering the relationship between the Reynolds number, the absolute pipe roughness, and the internal pipe diameter.

$$f = \frac{0.25}{\left[ \log \left( \frac{k_s}{3.7D} + \frac{5.74}{\mathrm{Re}^{0.9}} \right) \right]^2},$$ (8)

### 2.3. Unsteady Friction Model (UFM)

The emptying process exhibits a transient flow behavior, and then an unsteady friction model needs to be assessed. The model proposed by Brunnone [18,19] is considered to quantify the head losses per unit length considering the transient phenomenon:

$$J_u = J_s + \frac{k_\delta}{g} \left( \frac{\partial v}{\partial t} - a \frac{\partial v}{\partial s} \right),$$ (9)

where:

$J_u$: head losses per unit length in the unsteady flow regime;
$k_\delta$: Brunone friction coefficient;
$a$: wave speed, m s$^{-1}$;
$\partial v / \partial t$: local acceleration;
$\partial v / \partial s$: convective acceleration;
$s$: distance, m.

In particular, the rigid water column model neglects the convective acceleration term ($\partial v / \partial s = 0$); thus, the unsteady friction model is:

$$J_u = J_s + \frac{k_\delta}{g} \frac{dv}{dt},$$ (10)

The Brunnone friction coefficient is computed as:

$$k_\delta = \frac{\sqrt{C^*}}{2},$$ (11)

where:

$C^*$: Vardy's shear decay coefficient.

The Vardy's shear decay coefficient ($C^*$) depends of the type of flow regime [7]; for a laminar flow, it takes a value of 0.00476, and for a turbulent flow, it is calculated as:

$$C^* = \frac{7.41}{\mathrm{Re}^{\log_{10}(14.3/\mathrm{Re}^{0.05})}},$$ (12)

Taking into account Equation (10) and plugging it into Equation (1) can demonstrate a rigid water column model with a UFM:

$$\frac{dv}{dt} = \frac{\frac{p_1^* - p_{atm}^*}{\rho_w L} + g\frac{\Delta z}{L} - f\frac{v|v|}{2D} - \frac{R_v g A^2 v|v|}{L}}{1 + \frac{\sqrt{\frac{7.41}{\text{Re}^{\log(14.3/\text{Re}^{0.05})}}}}{2}},$$ (13)

The algebraic–differential equations system, composed by Equations (2), (3), and (13), represents the modeling of the condition of a transient flow for a draining process without admitting air with a UFM.

## 3. Experimental Stage and Numerical Runs

The experimental stage consists of a 4.36 m long methacrylate pipe with an internal diameter of 42 mm located at the Hydraulic Lab of the Universitat Politècnica de València, Valencia, Spain. The emptying process starts at rest with a ball valve at the closed downstream end. Figure 1 shows the configuration of the experimental facility. After this, the ball valve is suddenly opened and the water column begins to come out of the experimental pipe. The air phase starts to fill the hydraulic installation until the transient event finishes, because the opening percentage of the ball valve is not enough to generate the backflow air phenomenon and there is not an air valve to admit air into the installation. The experiments were conducted using a downward pipe slope of 0.457 rad. Table 1 shows the conducted numerical and experimental runs considering different initial air pocket sizes ($x_0$), and various resistance coefficients ($R_v$) of the ball valve. A pressure transducer was installed at the highest point of the installation to record the air pocket pressure. A detailed explanation of the experimental stage is presented in the publication conducted by Fuertes-Miquel et al. (2019) [3]. The diameter of the ball valve was 42 mm, and the opening percentages varied from 6% to 12%. The ball valve was characterized in the hydraulic lab of the Polytechnic University of Valencia thought measurements of absolute pressure and water flow. The experiments were performed with an air temperature of 26 °C. Air pocket sizes were measured using some marks in the pipe system. The surface tension and the viscosity were negligible in the mathematical model developed by the authors [22]. During all experiments, the sub-atmospheric pressure conditions did not reach the water vapor condition. The governing equations are valid for opening a percentage of the ball valve, where backflow air occurrence does not occur [1].

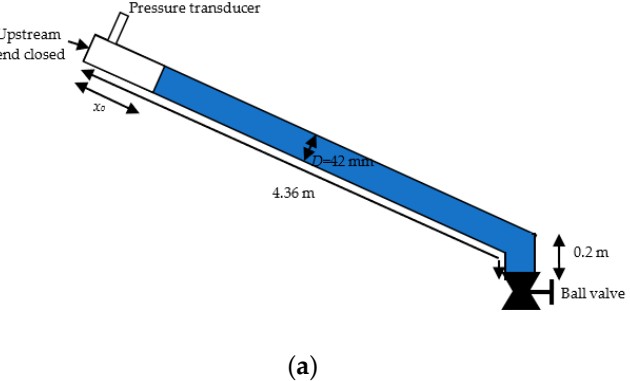

(**a**)

**Figure 1.** *Cont.*

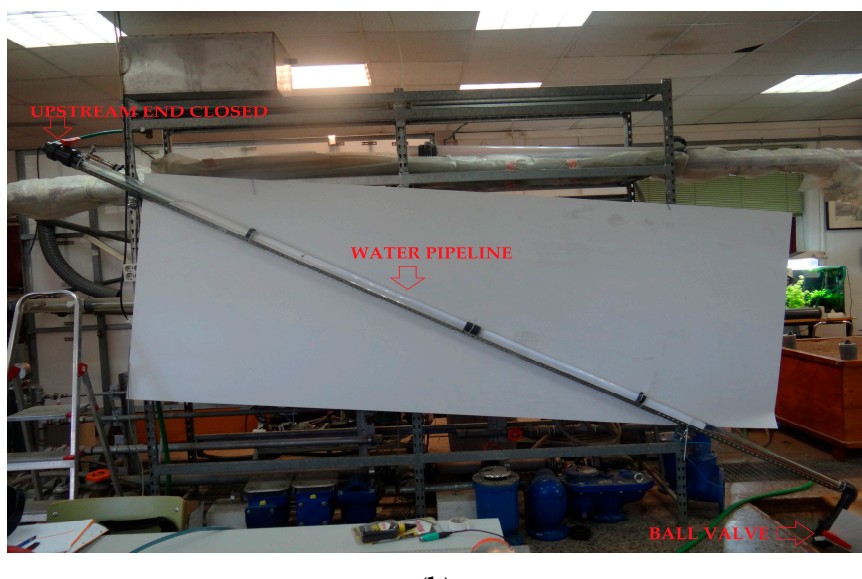

(**b**)

**Figure 1.** Configuration of the experimental facility: (**a**) Scheme of the experimental configuration; (**b**) photograph of the hydraulic installation.

**Table 1.** Initial conditions of the experimental and numerical runs.

| Run No. | $x_0$ (m) | $R_v \times 10^{-6}$ (s$^2$ m$^{-5}$) |
|---------|-----------|---------------------------------------|
| 1 | 0.205 | 11.89 |
| 2 | 0.340 | 11.89 |
| 3 | 0.450 | 11.89 |
| 4 | 0.205 | 25.00 |
| 5 | 0.340 | 22.68 |
| 6 | 0.450 | 30.86 |

The mathematical model developed by the authors [3,12] was applied for the initial conditions presented in Table 1. Equations (1)–(3) describe the behavior of the air pocket pressure, length of the water column, and water velocity that occur during the emptying process presented in Figure 1. The numerical runs were performed using a constant friction factor of 0.018 [3]. In this research, the friction factor was computed in the laminar zone as $f = 64/\text{Re}$, while different approaches were used to calculate the friction factor for the turbulent zone. A polytropic coefficient of 1.4 (adiabatic process) was utilized, since a rapid transient event occurs in this branch of the water pipeline. Considering a vertical pipe (see Figure 1), the gravity term is expressed as:

$$\frac{\Delta z}{L} = \frac{L - 0.2 \text{ m}}{L}sin(\theta) + \frac{0.2 \text{ m}}{L}cos(\theta), \tag{14}$$

Figure 2 shows the results of the air pocket head pressure patterns of the experimental and numerical runs. Figure 2a presents how the mathematical model developed by the authors is suitable for predicting the air pocket pressure patterns of Run No. 1 and No. 5, as shown with the comparison of the two experimental repetitions, which are statistically representative since the experimental measurements of air pocket pressure presented similar behaviors. According to the publication conducted by Fuertes-Miquel et al. (2019) [3], the mathematical model can adequately simulate runs from No. 1 to No. 6. Figure 2b shows the results of the numerical model for all runs. The minimum value of air pocket head pressure of 8.03 m was reached for Run No.1, while Run No. 6 exhibited the maximum value of air pocket pressure (8.46 m) compared to the other runs. At the ends of

the transient event, part of the length of the water column remained inside the hydraulic installation. In this sense, the air pocket pressure tended to a constant value.

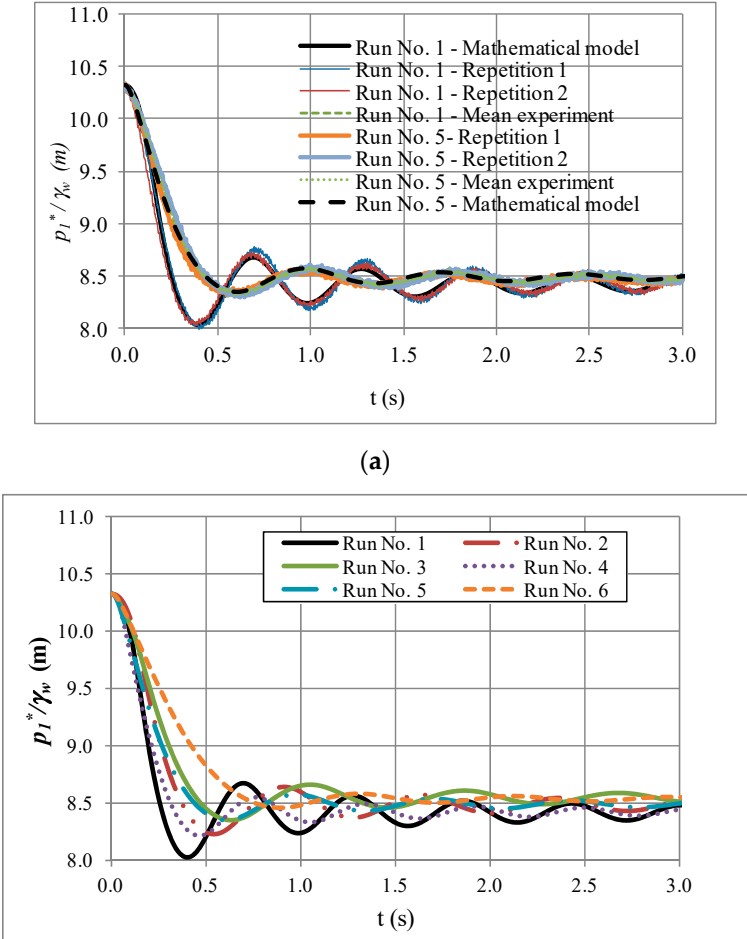

**Figure 2.** Air pocket head pressure patterns: (**a**) Comparison between the calculated and measured air pocket head pressure patterns for Run No. 1 and No. 5; (**b**) numerical runs from No. 1 to No. 6.

## 4. Results and Discussions

### 4.1. Steady Friction Model

This section shows the application of different formulations to compute the friction factor during the emptying process occurrence. The used formulations for computing the friction factor are described in Section 2.1. When the mathematical model was developed, a constant friction factor of 0.018 was used for calibration purposes. Thus, the Swamee–Jain, Hazen–Williams, Wood, and Moody formulations needed to be evaluated to compare numerical results. For the analysis, an absolute roughness pipe ($k_s$) of 0.0015 mm and a Hazen–Williams coefficient ($C_{CH}$) of 150 were used to represent the roughness characteristic of the methacrylate pipe.

The mathematical model was applied considering the formulations to compute the friction factor. Figure 3a shows the evolution for the friction factor during the transient event for Run No. 1. At the beginning of the transient flow, the friction factor took an asymptotic value since the system was at rest. The time from 0.049 to 0.350 s was characterized by a turbulent zone, where the Wood formulation presented the highest values, but they were compared to the Moody equation. The Swamee–Jain and Hazen–Williams equations achieved similar values during this time. From 0.39 s until the end of the hydraulic event exhibited the same behaviors, since a laminar flow was reached.

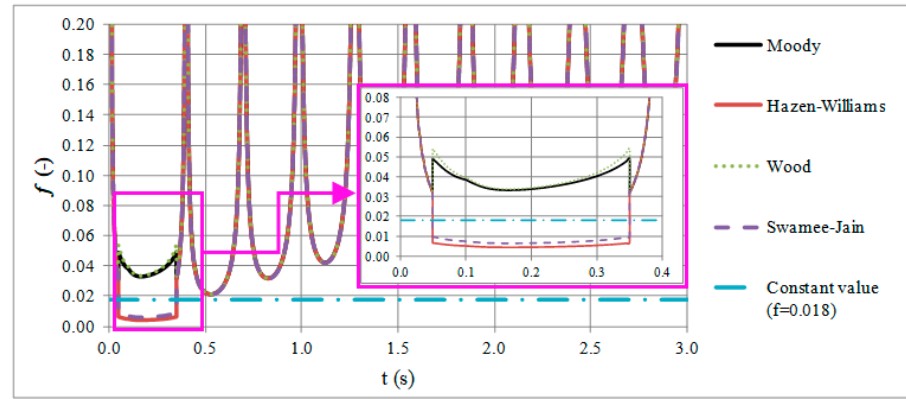

(a)

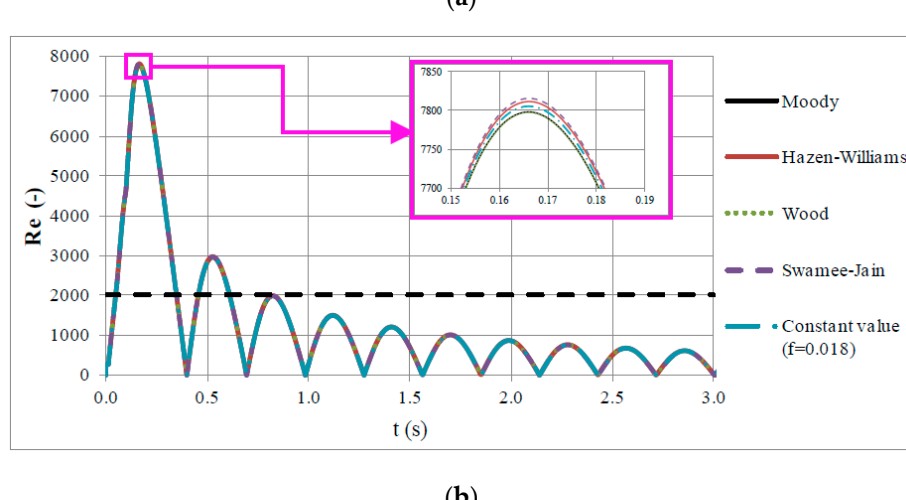

(b)

**Figure 3.** Behavior of the friction factor and Reynolds number for Run No. 1: (**a**) Friction factor and (**b**) Reynolds number.

Figure 3b shows that the Reynolds number pattern exhibited a similar behavior considering all friction factor formulations. The Swamee–Jain equation produced the maximum Reynolds number with a value of 7810, while the Moody equation provided the minimum value of 7797 during the entre hydraulic event.

Regarding the air pocket head pressure, the results confirm again that the use of friction factor formulations does not affect the patterns (see Figure 4a), since the same behaviors were practically obtained. Using a constant friction factor of 0.018, the mean pattern behavior was found. For Run No. 1 the minimum value of air pocket head pressure reached was 8.026 m, which is so close to minimum values of 8.027 and 8.025 m considering Moody and Hazen–Williams formulations, respectively.

The water velocity and the length of the water column are shown in Figure 4b,c, respectively. In both patterns, all friction factor formulations provided practically the same results regarding the oscillation patterns.

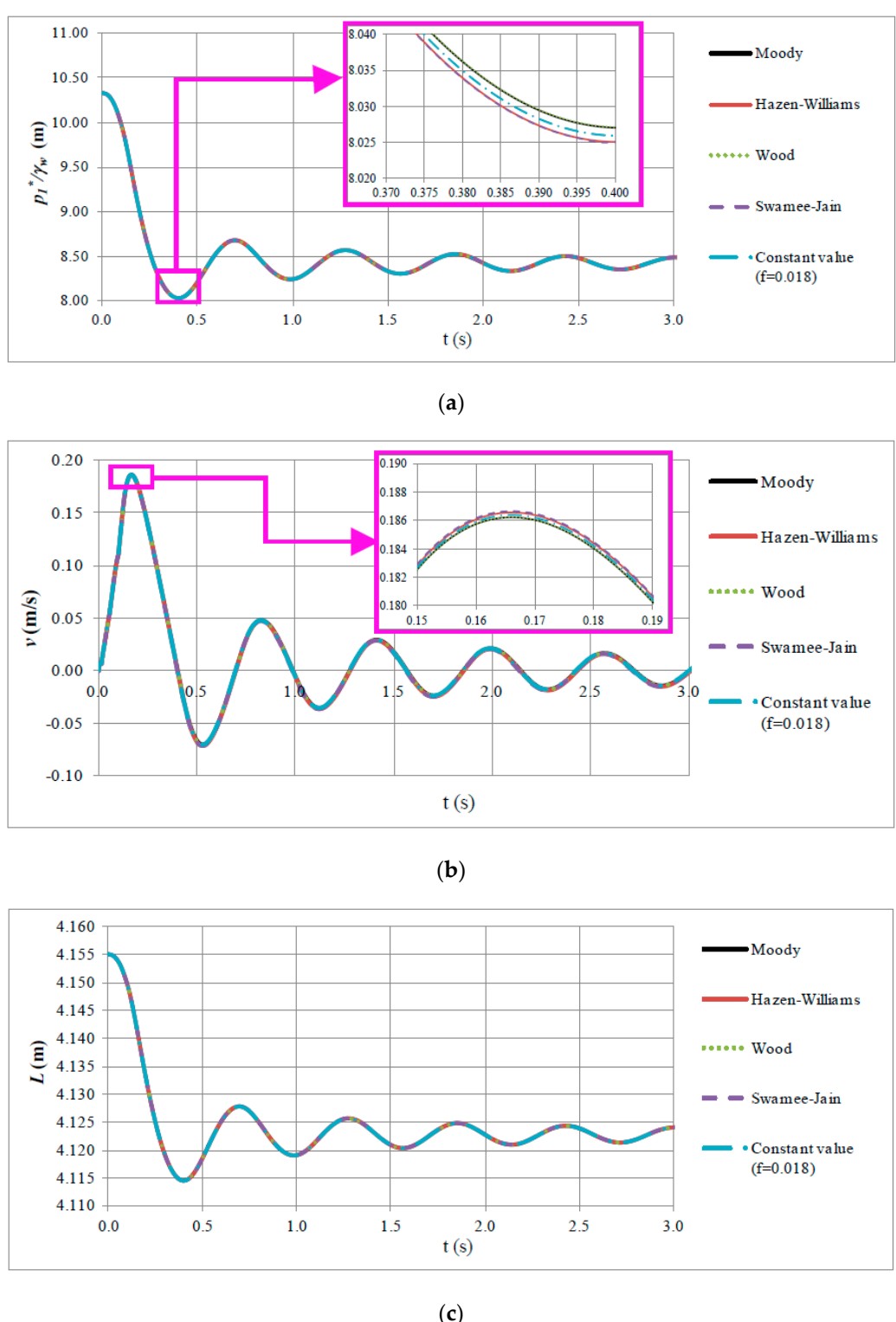

**Figure 4.** Evolution of the emptying process variables for Run No. 1: (**a**) Air pocket absolute pressure; (**b**) water velocity; (**c**) water column length.

### 4.2. Unsteady Friction Model

A comparison between the steady and unsteady friction models was conducted to note the order of magnitude generated by the local acceleration ($dv/dt$) on the behavior of the emptying process. Figure 5a presents the Reynolds number evolution for Run No. 5, where the turbulent flow condition was achieved from 0.015 to 0.465 s (see Figure 5a). For this run,

all friction factor formulations provided practically the same trend for the Reynolds number. Figure 5b contains the head losses per unit length using an SFM and a UFM. Peaks values of the head losses per unit length of 1.0715 mm/m ($J_u$) and 1.0706 mm/m ($J_s$) were reached at 1.155 s using the Wood formulation with the UFM and the SFM, respectively. The $J_s$ and $J_u$ minimum values were attained using the Hazen–Williams formulation. The constant friction value of 0.018 provides the mean behavior of the head losses per unit length.

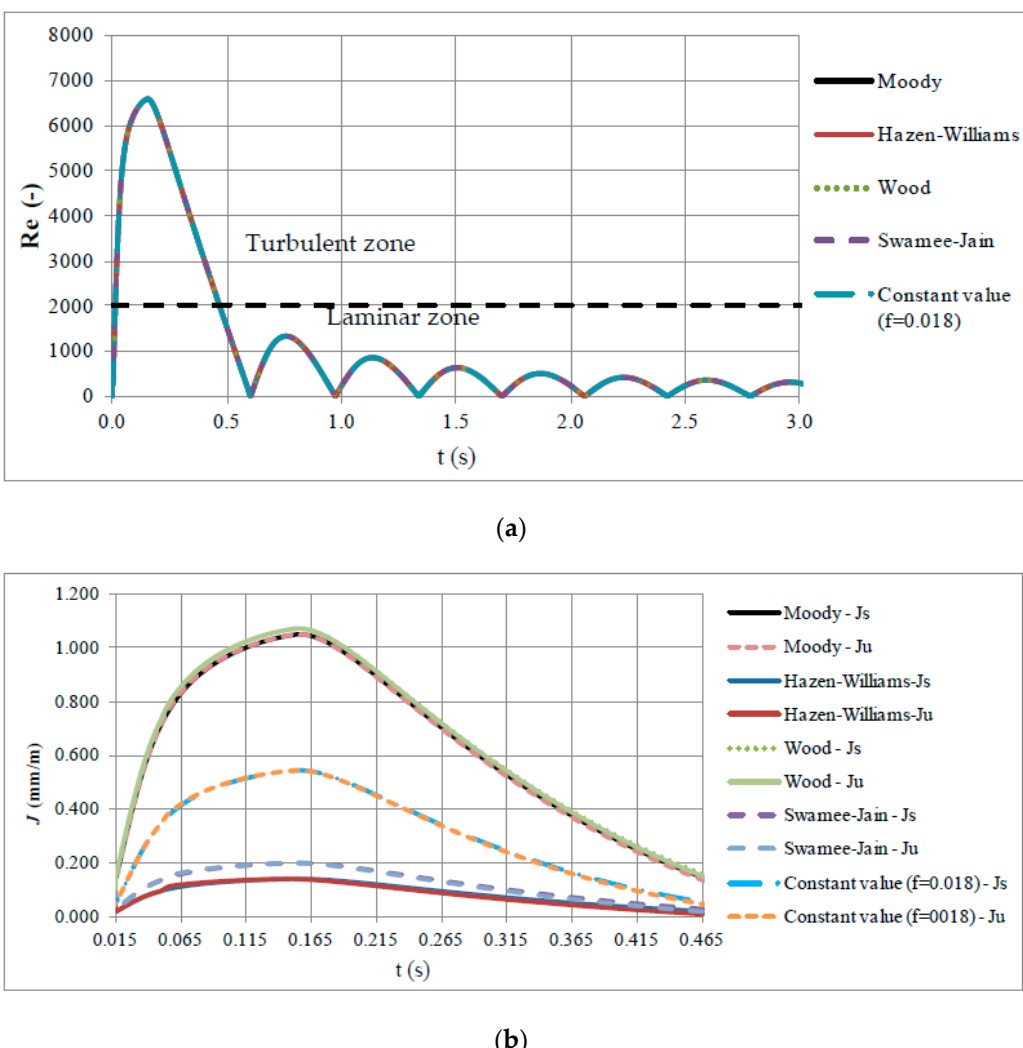

**Figure 5.** Analysis of Run No. 5: (**a**) Reynolds number evolution for an SFM condition; (**b**) behavior of the head losses per unit length.

A comparison between the SFM and the UFM using the Swamme–Jain equation was performed for Run No. 5 in the turbulent zone (from 0.015 to 0.465 s), since during this time, the greatest discrepancies were observed. Figure 6 shows the results of the air pocket pressure pattern, water velocity evolution, and water column length, where practically negligible discrepancies were found during the simulations.

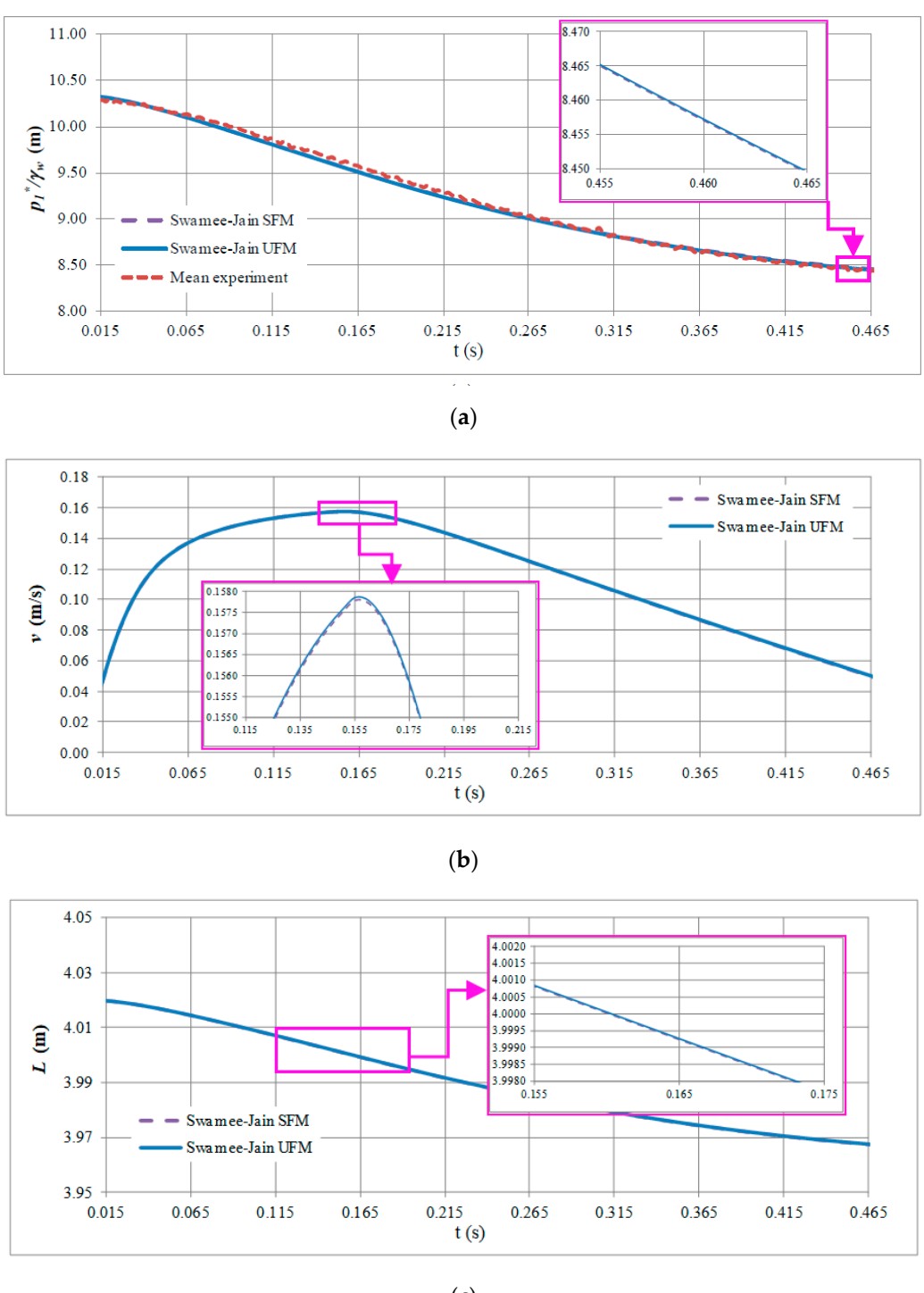

**Figure 6.** Comparison of the measured and computed patterns of Run No. 5 using the Swamee–Jain formulation with the SFM and the UFM: (**a**) Air pocket pressure; (**b**) water velocity; (**c**) water column length.

Finally, the root mean square error (*RMSE*) was computed in order to note the variation in the trends of the UFM and SFM formulations when computing the air pocket pressure values.

$$RMSE = \sqrt{\frac{1}{n} \sum_{i=i}^{n} \left[ \frac{p^*_{1,exp} - p^*_{1,num}}{p^*_{1,exp}} \right]^2}, \tag{15}$$

where subscripts *exp* and *num* represent the mean experimental values (see Figure 6a) and the numerical simulation of air pocket pressure, respectively, and *n* is the data size.

The best fits were achieved with the UFM using the Moody and Wood formulations, with an RMSE value of 0.367%, while and the worst fits were obtained using the UFM with the Hazen–Williams and Swamee–Jain formulations with values of 0.376% and 0.375%, respectively. However, taking into account the order of magnitude of the RSME (from 0.369% to 0.375%), it is important to mention that there were practically no discrepancies using the different formulations of the friction factor equation, or the SFM and UFM.

## 5. Conclusions and Recommendations

The draining process was analyzed considering steady and unsteady friction models in order to note the effect on the accuracy of the current one-dimensional mathematical models for simulating this operation. The numerical and experimental results of air pocket pressure patterns were compared in a pipeline with a length of 4.36 m and an internal diameter of 42 mm to compute the values using a UFM and an SFM. Based on the results, the following conclusions can be drawn:

- During the emptying process, the air pocket pressure started under atmospheric conditions. When the ball valve located at downstream end was opened, the absolute pressure pattern descended until the lowest value (first drop), after which some oscillations were reached until the water column was again at rest. The length of the water column showed a similar behavior. Regarding the water velocity, it started at rest (0 m s$^{-1}$), following which it rapidly reached the maximum value, and finally negative and positive values were generated.
- Considering the six experimental runs, the implementation of the unsteady friction model of Brunone in the simulation of the draining process better fixed the measured air pocket pressure oscillations in the analyzed experimental facility. When the Moody and Wood formulations were implemented with the UFM, the minimum root mean square errors were reached.
- It is important to highlight that both the SFM and the UFM adequately predicted the air pocket pressure oscillations using all of the empirical formulations to compute the friction factor.
- The mathematical model proposed considers the analysis of the laminar and turbulent zone flows. The first drop of sub-atmospheric pressure pattern is the more complex zone to simulate, since it involves the presence of laminar and turbulent flows. After that, the water movement is almost null; consequently, the laminar flow is presented during this part of the transient event.

**Author Contributions:** Conceptualization, Ó.E.C.-H. and V.S.F.-M.; methodology, J.R.C.-H.; software, G.G. and I.D.; validation, Ó.E.C.-H.; orginal draft, Ó.E.C.-H. and V.S.F.-M.; and final review, J.R.C.-H., G.G., and I.D. All authors have read and agreed to the published version of the manuscript.

**Funding:** This research was supported by the Research Department of the Universidad Andres Bello, grant number DI-12-20/REG; and the University of Santiago of Chile with DICYT-Project Number 062117DC.

**Institutional Review Board Statement:** Not applicable.

**Informed Consent Statement:** Not applicable.

**Data Availability Statement:** Not applicable.

**Conflicts of Interest:** The authors declare no conflict of interest.

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
