# Peer review of "Assessment of Steady and Unsteady Friction Models in the Draining Processes of Hydraulic Installations"

_water, doi:10.3390/w13141888_

Round 1

Reviewer 1 Report

See uploaded file.

Author Response

Dear reviewer 1:

Please see enclosed file

Reviewer 2 Report

Assessing of steady and unsteady friction models in draining processes of hydraulic installations.

WATER - 1249132

Comments Comments and Suggestions for Authors

A brief summary

The manuscript is designed to determine the hydraulic characteristics in the conditions of water outflow from a pipe with a pillow - an air pocket. A section filled with air is then formed at the highest point of the conduit. However, there is a supply line in piping systems, which the authors of the article did not take into account. Laboratory tests of the hose indicate that it is only a part falling down with a bottom valve. The air collects in the upper part of this section. The Authors described the flow resistance with the coefficients of linear hydraulic losses along the length of the conduit. The described hydraulic scheme is the emptying of the vessel / hose with an air cushion. The authors modelled the cyclical process of decaying air supply to a confined space.

Broad comments and specific comments

  1. Introduction

  1. Mathematical model

(R. 81) It should be. 2.1. Governing equations

(R.90). Methodical error. Must be corrected.

 - atmospheric pressure, 101325 Pa.

How the atmospheric pressure was kept constant throughout all tests. This is impossible under laboratory conditions (see Figure 1, Figure 2 for X=0, Figure 4a for X=0).

Comment.

This is normal atmospheric pressure at sea level, in normal air temperature. The pressure and temperature in the laboratory were current (up-to-date). Atmospheric pressure varied with each test. The atmospheric pressure was different at the start of the test and at the end of the test. For a short time interval of a single test, the atmospheric pressure can be defined as constant. The pressure of water vapour also depends on the atmospheric pressure and temperature. The water vapour pressure affects the effectiveness of the reducing orifice-nozzle (short thin pipe) stop valves (with minimum flow diameter). Temperature affects the specific density of water and air and the strength of the surface tension of the water.

(R.92)

Comment.

Please explain (shortly). How was the length of the water column L (m) measured during the test period? The initial length of the water column was Lo (m). The end length of the water column was asymptotically ....... (m).

Eq. 1 - 13 (in all). It should be corrected.

Comment.

Why do equations end with a comma. The comma at the end of the equations must be removed.

Comment.

Descriptions of variables for formulas should be reformatted.

(R.87-98) Eq.1.

For example, Eq. 1:

where: v – water velocity (ms-1), air pocket pressure (Pa),  – atmospheric pressure (Pa), et cetera.

(R.105-108) Eq.3.

(R.119) Eq.4.

(R.130-131) Eq.5.

(R.139-140) Eq.7.

(R.151-157) Eq.9.

(R.161-162) Eq.11.

Eq. 6. It should be corrected.

Comment.

There is not the dependent variable.

(R. 112-114) It should be corrected.

Comment.

There is no description of the variable Lt - - up to this point in the text.

(R. 116) It should be. 2.2. Steady friction models (SFM)

(R. 146) It should be. 2.3 Unsteady friction models (UFM)

  1. Experimental stage and numerical runs

Comment.

There is no information about the geometry of the lower valve. In the opinion of the Reviewer, the diameter of the bottom valve opening was very small. The bore diameter of the lower valve is decisive here. The described phenomenon will not occur when this diameter is greater than the limiting value of the valve diameter. It is related to the equilibrium of the strength of the surface tension of the water.

(R. 191) - Table 1. It should be corrected.

Comment.

In engineering practice, we use valve opening parameters. In calculations we used various resistance coefficient (??) of the ball valve. However, the investigated phenomenon depends on the area of the active opening of the valve. Please add in Table 1 columns:

The current atmospheric pressure

The current air temperature

The current water temperature

Valve opening degree

Areas of the active opening of the gate valve

Ratio of the valve opening area to the conduit cross-sectional area

Start run

End run

1

2

3

4

5

6

(R. 191) - Table 1. (R.201-202) - text. Figure 2a. It should be corrected.

Comment.

Table 1. Initial conditions of experimental and numerical runs, was 5 runs. In text and on the Figure 2a there are 4 repetitions (experimental). In Reviver opinion the all runs were made in repetitions (statistically minimum 3, it depends of measurement uncertainties). In all runs used mean results experiment. On the Figure 2a were are the 4 repetitions and 2 mean experiments. The repetition is the basic methodology of laboratory experiments. Please corrected the status of experiments (mean or simple run). The analyzed results are to be homogeneous - its should be consistent with Table 1.

(R. 208-209). Figure 2. It should be corrected.

Comment.

The chart legend should have a white background and a black outline.

  1. Results and Discussions

Comment.

The Authors do not discuss the procedure for calculating the hydraulic parameters of the model used. They only compare the results of selected tests. In the opinion of the Reviewer, in this part the article gives the impression of a "black box".

(R. 250-251). Figure 4. It should be corrected.

It should be.

: a – air pocket absolute pressure; b – water velocity; c – water column length.

  1. Conclusions and recommendations

(R. 293). 5. Conclusions and recommendations. It should be corrected.

Comment.

The conclusions should take into account the obtained results.

Author Response

Dear Reviewer 2

Please see enclosed file
